# Analysis of a 180-degree U-turn maneuver executed by a hipposiderid bat

**Peter Windes**, **Danesh K. Tafti***, **Rolf Müller**

Department of Mechanical Engineering, Virginia Tech, Blacksburg, VA, United States of America

* dtafti@exchange.vt.edu

## Abstract

Bats possess wings comprised of a flexible membrane and a jointed skeletal structure allowing them to execute complex flight maneuvers such as rapid tight turns. The extent of a bat's tight turning capability can be explored by analyzing a 180-degree U-turn. Prior studies have investigated more subtle flight maneuvers, but the kinematic and aerodynamic mechanisms of a U-turn have not been characterized. In this work, we use 3D optical motion capture and aerodynamic simulations to investigate a U-turn maneuver executed by a great roundleaf bat (*Hipposideros armiger*: mass = 55 g, span = 51 cm). The bat was observed to decrease its flight velocity and gain approximately 20 cm of altitude entering the U-turn. By lowering its velocity from 2.0 m/s to 0.5 m/s, the centripetal force requirement to execute a tight turn was substantially reduced. Centripetal force was generated by tilting the lift force vector laterally through banking. During the initiation of the U-turn, the bank angle increased from 20 degrees to 40 degrees. During the initiation and persisting throughout the U-turn, the flap amplitude of the right wing (inside of the turn) increased relative to the left wing. In addition, the right wing moved more laterally closer to the centerline of the body during the end of the downstroke and the beginning of the upstroke compared to the left wing. Reorientation of the body into the turn happened prior to a change in the flight path of the bat. Once the bat entered the U-turn and the bank angle increased, the change in flight path of the bat began to change rapidly as the bat negotiated the apex of the turn. During this phase of the turn, the minimum radius of curvature of the bat was 5.5 cm. During the egress of the turn, the bat accelerated and expended stored potential energy by descending. The cycle averaged total power expenditure by the bat during the six wingbeat cycle U-turn maneuver was 0.51 W which was approximately 40% above the power expenditure calculated for a nominally straight flight by the same bat. Future work on the topic of bat maneuverability may investigate a broader array of maneuvering flights characterizing the commonalities and differences across flights. In addition, the interplay between aerodynamic moments and inertial moments are of interest in order to more robustly characterize maneuvering mechanisms.

## 1. Introduction

Of the three animal groups capable of powered flight—insects, birds, and bats—bats uniquely possess highly flexible wings composed of a jointed skeletal structure and a thin compliant

**Data Availability Statement:** All relevant data are within the manuscript and its Supporting Information files.

**Funding:** This research received financial support from NSF CBET Grant No. 1510797, NSF IRES

Grant No. 1658620, support from VT ICTAS/BIST Center, National Natural Science Foundation of China (Grant Nos. 11374192 & 11574183), and Chinese Ministry of Education Tese Grant for international faculty exchange.

**Competing interests:** The authors have declared that no competing interests exist.

skin membrane with embedded muscles. This wing morphology allows a bat to perform complex flight maneuvers which are critical to navigating cluttered environments and capturing prey. Three fundamental factors in understanding the maneuvering flight of bats are the shape and structure of their wings (morphology), the motion of the wings during flight (kinematics), and the interaction of the wings with the surrounding air (aerodynamics).

Early work on characterizing the maneuverability of bats focused on wing morphology. In 1987, Norberg and Rayner investigated the relationship between bat wing morphology and flight capabilities such as maneuverability [1]. They studied 257 species of bats from 16 of the 18 families using multivariate statistical correlations in order to link morphological parameters to flight performance. Specifically, they investigated the relationship between mass, wing loading, wing aspect ratio, and wingtip shape index with flight efficiency, maximum flight speed, maneuverability, agility, and load carrying capability. It was found that across bat species, variations in morphology were closely correlated with optimization of different flight capabilities. Further, it was hypothesized that differing feeding strategies such as hunting for insects among vegetation, perching while seeking insect prey, fish catching, or foraging for fruit were a key driver of optimization of different flight abilities in different species of bats. In their analysis, low wing loading was correlated with increased maneuverability. While the scope of their study was limited to bats, it has been noted that bats tend to have lower wing loading than birds and also tend to be more maneuverable [1, 2]. This study represented a material advance in the understanding of the factors that impact flight performance of bats including maneuverability; however, a limitation of the study was that the specific mechanics of bat flight were not investigated. That is, morphological parameters were correlated with flight abilities, but the causal mechanism of the relationship remained unknown. Additionally, a portion of the analysis such as the investigation of flight efficiency relied on mathematical models based on fixed wing theory.

An early description of the turning radius of a flying animal as a function of mass, wing area, and bank angle was provided by Pennycuick [3]. However, two important factors in applying this framework had not been characterized by researchers—specifically, how do unsteady mechanisms impact lift generation in flying animals, and to what degree is banking utilized in turning flight. Banking during both fixed wing flight and flapping flight tilts the lift force laterally providing a centripetal acceleration. Thus, both the bank angle and the magnitude of the lift force impact turning mechanics. It was long hypothesized that unsteady aerodynamic mechanisms may be an important aspect of lift generation in flapping flight, but the degree of impact was not known [4]. Unsteady mechanisms were first quantified in insect flight [5, 6], and subsequently bat flight [7, 8]. Since then, certain unsteady mechanisms such as enhanced lift by the leading edge vortex (LEV) have been shown to be nearly ubiquitous in flapping flight, while other mechanisms such as clap and fling are only observed in certain insects operating at smaller length scales. Bats have also been shown to enhance lift by dynamically changing their wingspan [9].

In 1986, Aldridge studied the turning flight of six bat species—*Rhinolophus ferrumequinum*, *R. hipposideros*, *Plecotus auritus*, *Myotis mystacinus*, *M. daubentoni*, and *Pipistrellus pipistrellus* [10]. All species were relatively small with masses ranging from 6 g to 22 g, and wing spans ranging from 21 cm to 36 cm. Each bat was trained to execute a U-turn in an open-ended flight tunnel, and 3D reconstruction of the body trajectories was conducted allowing calculation of the flight velocity and the minimum radius of curvature during each maneuver. All the bats were observed to decelerate significantly during the turn to a minimum flight velocity of 0.29–0.71 m/s depending on the species. Additionally, they were observed to climb as they entered the turn and subsequently descend while existing the turn. The bank angles during each turn were between 50 and 90 degrees. A lift coefficient for each fight was estimated using

Pennycuick's formula. Considering variation between the bat species, Aldridge determined that the minimum radius of curvature was correlated with the mass of bat. Correlation was found between wing loading and curvature only when excluding results from *R. ferrmequinum*, and it was hypothesized that this bat was using a fundamentally different turning mechanism from the others. Due to the lack of specific kinematic and aerodynamic data provided, it is unclear what exactly this difference was.

While banking is the most obvious turning mechanism, it is not the only available method to achieve a turn during flapping flight. Fundamentally, some force asymmetry is required to turn; however, this may be achieved using either a thrust/drag asymmetry (yawing turn) or a lift asymmetry (banking turn). Iriarte-Diaz and Swartz studied the kinematics of a fruit bat (*Cynopterus brachyotis*) executing 90 degree turning maneuvers in an L-shaped flight tunnel [11]. They found that the use of a banking mechanism was significant but did not fully explain the turn. Heading rotation (change in body orientation) was observed to precede change in bearing (change in flight direction), and occurred more significantly during the upstroke. Since aerodynamic force was not calculated, they hypothesized that either elevated thrust on the outer wing or increased drag on the inner wing may have been employed to achieve a change in heading.

Henningsson et al. [12] conducted kinematic and aerodynamic analysis of a brown long-eared bat (*Plecotus auritus*) performing a basic sideways maneuver using particle image velocimetry (PIV) and 3D motion capture in a wing tunnel. Although due to limitations of the wind tunnel setup that constrained the magnitude of the heading change during the maneuver to 4 degrees, some interesting observations were realized. It was observed that the bats were adaptable in selecting their turning mechanism, using both the upstroke and downstroke to initiate the turn. Additionally, both lift as well as thrust/drag asymmetries were employed. The most common method for generating a force asymmetry was in thrust/drag during the upstroke and was identified in 7 of the 10 observed maneuvers. In this study, the ability to simultaneously analyze the wing kinematics along with aerodynamic forces was pivotal towards understanding the maneuvers.

We previously investigated (Windes et al. [13]) the kinematics and aerodynamics of a sweeping turn maneuver of a great round-leaf bat (*Hipposideros armiger*) using 3D motion capture and aerodynamic simulations. Using the kinematic data along with numerical airflow simulations, we were able to couple the kinematic and aerodynamic analysis to show the bat using synergistic yawing and banking mechanisms. The initiation of the turn was dominated by yawing generated by elevated thrust on the outer wing during both the upstroke and the downstroke. As the bat progressed through the turn, the banking mechanism became much more dominant and generated a large lateral force imparting a centripetal acceleration.

These four studies on maneuvering bat flight—Aldridge [10], Iriarte-Diaz and Swartz [11], Henningsson et al. [12], and Windes et al. [13]—are challenging to unify due to significant differences in the maneuvers, size of the bats, as well as different data collection approaches. Limited inference on turning mechanisms can be derived from Aldridge [10] since detailed kinematics were not collected and no aerodynamic measurements were taken. However, the data reported on flight velocity, radius of curvature, and bank angle may provide some general insight on the execution of a tight 180-degree maneuver over a range of bat species. Iriarte-Diaz and Swartz [11], Henningsson et al. [12], and Windes et al. [13] all suggest that yawing in addition to banking is utilized across a diverse range of maneuvers from 4˚ to 90˚ heading change. Further, the upstroke was universally found to play at least some roll in executing a maneuver in contrast to earlier notions that banking alone was used. The mode of turning analyzed by Iriarte-Diaz and Swartz and Windes et al., were similar and comparable mechanisms were observed in both studies—specifically the bat employed a combined yawing and banking

mechanism with an emphasis on the upstroke. However, exact comparison between these two studies is challenging since aerodynamic data was not provided by Iriarte-Diaz and Swartz. Henningsson et al. observed both elevated thrust on the outer wing as well as increased drag on the inner wing, while Windes et al., only observed elevated thrust on the outer wing. Henningsson et al. also observed initiation of a maneuver using a banking mechanism.

Additional generalizations between the studies are unclear, and perhaps not appropriate. It is likely that while commonalities exist in the mechanisms used by different bats to maneuver, every maneuver will not be executed in the same way. From the limited data in the literature on the topic of maneuvering bat flight, early indications are that similar turning maneuvers across bat species share more commonalities than different maneuvers executed by bats within the same species. That is, the specific turning mechanisms used will be quite dependent on what the bat is trying to achieve, but for similar maneuvers commonalities may be observed across different bat species.

In trying to determine the limits of maneuverability of a bat, studying a tight U-turn can provide important insight. More gradual turns by definition do not represent the maximum possible turning capability of a bat. The data reported by Aldridge [10] particularly illustrates the remarkable degree of maneuverability achievable by a bat. For example, *R. ferrumequinum* executed a U-turn with a minimum radius of curvature less than 1 cm. In the present work, we turn our attention to studying a similar type of turn executed by an *H. armiger* with the benefit of detailed kinematic and aerodynamic data collection and analysis tools. It is of particular interest to investigate any similarities or differences between the mechanisms used to execute a gradual turn versus a tight turn. During this analysis, we will address several key questions. What is the difference in wing kinematics between the right and left wings during the U-turn maneuver? How do the wing kinematics during a U-turn differ from straight flight? Do the wing kinematics during a U-turn primarily cause lift asymmetries, thrust asymmetries, or both? Is the U-turn achieved through banking, yaw rotation, or a combination of both? What is the relative contribution of the upstroke and downstroke during the U-turn? Comparing a sweeping turn with a U-turn, do the respective mechanisms differ in degree or in kind? What is the relative energy cost of a U-turn compared to straight flight or a sweeping turn?

## 2. Methods

Measurements of wing kinematic data of a great roundleaf bat (*Hipposideros armiger*) were taken inside a flight tunnel using an optical motion capture system of 28 synchronized cameras (GoPro Hero 4 Black). In order to aid wing tracking, 240 small white markers were affixed to the bat's wings. Video recording was conducted at a frame rate of 120 Hz and in $1920 \times 1280$ resolution. The large number of cameras in the setup served to both reduce self-occlusion of the wings by capturing more perspectives of the bat, as well as enlarging the capture volume. A minimum of two cameras is needed to observe a particular point in order to generate a 3D representation of that point; however, camera observation redundancy increases the robustness and precision of the measurements. The calibration of the cameras was preformed using the open source Svoboda Multi-Camera Self-Calibration toolbox [14].

The bat was collected from a cave in southern China. Ethical bat housing and handling guidelines were observed, and approval for the study was granted by Virginia Tech's Institutional Animal Care and Use Committee (IACUC protocol number 15–067). While not in the flight tunnel, the bat was housed in a temperature and humidity controlled aviary. After conclusion of this study, the bat was housed for further research.

During flight data collection, bat was allowed to fly freely inside the 1.2 m × 1.2 m tunnel, and a 180˚ U-turn flight was selected for analysis. Videos of the flight from each camera were

processed using an in house MATLAB code to generate a time series of the 3D coordinates of the wings. The code applies standard methods of stereo triangulation coupled with a series of bat flight specific predictive motion models to aid the user in generating point correspondences. The most challenging and labor intensive aspect of collecting high spatial resolution bat wing kinematic data is establishing corresponding points between different cameras as well as between the time series of frames—referred to as the point "correspondence problem." Many of the automated methods which exist for automatically generating correspondences do not apply well to bat flight since bat wings are highly flexible and often twist and bend in unpredictable ways. Based on a survey of the literature most researches report using manual methods to establish correspondences when collecting bat flight kinematic data.

Since our aerodynamic analysis was conducted using computational simulations which require a relatively dense constellation of points to capture the wing kinematics, the use of purely manual methods for video processing was untenable. In order to address these challenges, we employ a hybrid user-supervised pseudo automated method. A fully automated correspondence method is attractive in principle but in practice introduces some problems with data quality. Since we do not have *a priori* knowledge of the correct kinematics, there is no reliable way of verifying automatically identified correspondences. Manually checking the results of fully automated tracking becomes prohibitively labor intensive—for example, in the present work a time series of 90 frames captured by 28 cameras contains 2,520 discrete views collectively containing $(240) \times (2,520) = 604,800$ image points to verify. In contrast, the user-supervised pseudo automated method used in the present work leverages a number of tools to maintain data fidelity while being able to process data considerably faster than a purely manual method.

After collection and 3D reconstruction of the kinematic data, we conducted numerical simulations of the airflow around the bat's wings which allow for the calculation of aerodynamic forces, moments, and power expenditure. The simulations were conducted using an incompressible Navier-Stokes solver [15] with the immersed boundary method (IBM) [16] to represent the wings in the flow. The IBM consists of embedding a surface mesh at the location of the bat wing inside the fluid domain and moving it at each integration time step as dictated by the wing kinematic data. During solution of Navier-Stoke, a no-slip boundary condition is imposed at the location of the embedded wing mesh on both the top and bottom surface of the mesh. Further details on the implementation of the method are given in Windes et al. [17, 18].

In the present study, a structured Cartesian fluid grid containing 69.1 million cells was used for the simulation. In the proximity of the bat, the fluid grid was refined to approximately 44 cells per wing chord length based on our prior grid independence analysis for similar simulations of bat flight [17, 19]. Validation of the computational method including selection of the grid refinement level is discussed further in the results section.

During the simulation, the fluid domain was given a fixed reference frame velocity equal to the mean flight velocity of the bat to minimize the net displacement of the bat inside the domain. The velocity field was initialized at the reference frame velocity. Since the bat performed a 180˚ U-turn partially returning toward the starting location, the net displacement was primarily a result of the initial 2–3 wingbeat cycles leading into the turn (see Fig 2A). In this context, the motion of the bat surface mesh relative to the background fluid grid represents the perturbation of the body location from the mean trajectory. The actual flight trajectory is recovered in post-processing by adding back the reference frame velocity to the perturbation. Since the bat travels back towards its initial position after the U-turn, there is less opportunity to constrain the size of the fluid domain by substantially moving the reference frame. Consequently, the computational cost of a U-turn simulation is larger than a straight flight of comparable duration.

The temporal discretization in the simulation was 20 microseconds per integration time step. Since this is significantly smaller than the video sample rate, intermediate point locations in the kinematic data were interpolated using piecewise cubic splines. The simulation was parallelized across 120 CPU cores using distributed memory message passing interface (MPI). The full simulation required approximately 44,000 CPU-hours to complete and was run on Intel Xeon E5-2680v3 2.5 GHz hardware.

In post-processing, two reference frames are used throughout the analysis—a ground fixed coordinate system and a body fixed coordinate system which rotates and translates along with the bat's body. The ground frame was defined such that $z_g$ points upward exactly opposite to gravity and $x_g$ points along the length of the flight tunnel. The body fixed frame was defined such that the origin follows the approximate center of mass of the bat body and the $x_b$ direction remains aligned with the long axis of the bat's body and points forward. These coordinate systems are depicted in Fig 1.

When analyzing the wing kinematic data of a maneuvering flight, the right and left wings must be treated separately to identify asymmetries which drive the maneuver. A separate stroke plane is defined for each wing by joining the shoulder point to a regression line passing through the locus of points traced out by the wingtip (Fig 1). This calculation is done on the wingtip trajectory in the body fixed coordinate system. Since the stroke plane may vary over time, a new stroke plane is calculated for each cycle updated every half cycle. A vertical stroke plane angle is defined by projecting the wingtip regression line onto the body-fixed $x_b$–$z_b$ vertical plane and is the angle between the line and the $x_b$ axis. It is representative of the forward-backward movement of the wing–smaller the vertical stroke plane angle, larger is the fore-backward movement of the wing with respect to the flight direction. A horizontal stroke plane angle is defined by projecting the wingtip locus regression line onto the body-fixed $x_b$–$y_b$ plane and the angle between the projected regression line and the $y_b$ axis is taken as the horizontal stroke plane angle. It is representative of the lateral movement of the wingtip. Motion of the wings relative to the stroke plane is quantified by the flap angle, stroke plane deviation, and half span. The flap angle represents in plane rotation of the spanline while stroke plane

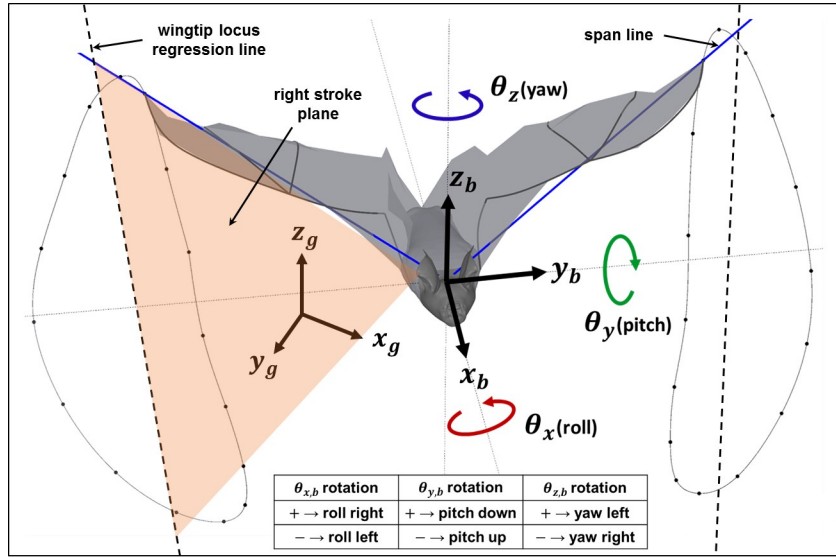

**Fig 1. Ground and body coordinate systems.** Rotations relative to the body frame are defined as roll, pitch, and yaw as depicted. A stroke plane is defined for the right and left wings separately by connecting the shoulder point with a regression line passing through the locus of wingtip points.

deviation represents out of plane rotation of the span line. The half span is the instantaneous distance between the shoulder and the wingtip.

The aerodynamic metrics calculated are force, moment, and power. The pressure and velocity fields in proximity to the wing surface which are generated by the simulation are used to calculate force on each discrete element of the wing ($\vec{F}_e$) over the duration of the flight. Net aerodynamic force represents the surface integration of the surface element forces. The rotational moment is calculated relative to the middle of the bat's body by integrating $\vec{M}_b = \vec{r}_e \times \vec{F}_e$ over the surface of the wings. Aerodynamic power was obtained by integrating $P_{aero} = \vec{F}_e \cdot \vec{v}_e$ over the wing surface. Total power expenditure was calculated as the sum of aerodynamic power, rate of change of kinetic energy, and rate of change of potential energy.

## 3. Results

The bat studied was an adult male *H. armiger* weighing 54.5 g and with a maximum outstretched wingspan of 51 cm as measured during the downstroke from the kinematic data. Relevant morphological parameters are shown in Table 1.

*H. armiger* is a somewhat larger bat in both mass and wing span compared to species previously investigated in prior studies of maneuvering bat flight [11, 12]. According to data reported by Norberg [1], the majority of bat species fall between 5 and 100 g with extreme values ranging from 2 g up to 1 kg. Based on a correlation of body mass and wing area over 257 bat species, the wing surface area of *H. armiger* is slightly larger than other species of similar mass. The aspect ratio of *H. armiger* falls near the middle of the distribution. The kinematic and aerodynamic mechanisms driving a flight maneuver can be compared across bat species; however, one must consider the morphological differences between the bats when making the comparison.

### 3.1 General description of U-turn flight

The flight selected for the present analysis consists of six wingbeat cycles over a duration of 690 ms and includes both the approach to the U-turn in addition to the U-turn itself. The Reynolds number of the flight defined with respect to the initial approach velocity and the mean chord is 10,000. Fig 2 shows the flight trajectory of the bat's body from the top view ($x_g$–$y_g$ projection) and the side view ($x_g$–$z_g$ projection). Over the course of the entire flight, the bat first begins to gradually turn and climb, then executes a tight U-turn, and lastly descends while exiting the turn. This sequence can be categorized into four phases:

- Phase 1, approach: Prior to entering the U-turn, the bat executes a gradual right turn, decreases its flight velocity, and begins to climb. This phase consists of wingbeat cycles 1 and 2.

- Phase 2, initiation of U-turn: The bat rapidly tightens the radius of the turn while continuing to climb and decelerate. This phase consists of wingbeat cycles 3 and 4.

- Phase 3, apex of U-turn: During the apex of the U-turn, the bat rapidly changes both heading and bearing angle. The bat reaches a maximum height, minimum radius of turn, and minimum flight velocity during this time. This phase consists of wingbeat cycle 5.

**Table 1. Morphological parameters of the subject.**

| Mass (g) | Span (cm) | Wing Area (cm²) | Planform area (cm²) | Mean chord (cm) | AR | Wing loading (N/m²) |
|---|---|---|---|---|---|---|
| 54.5 | 51 | 434 | 398 | 7.8 | 6.5 | 13.4 |

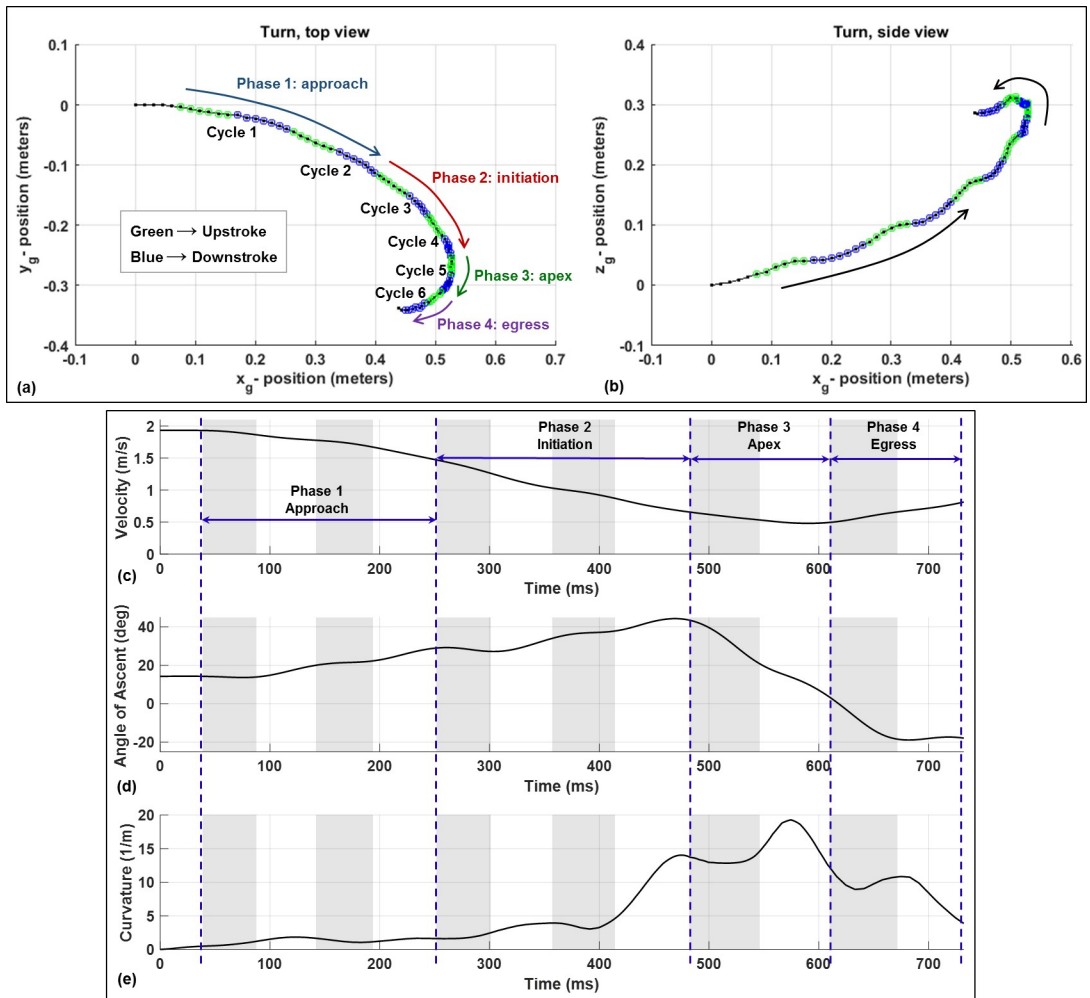

**Fig 2. Top: Body trajectory of the bat during the U-turn flight from the top and side view.** Bottom: Velocity, angle of ascent, and curvature of the turn. The curvature is defined using the standard definition: (radius of curvature)$^{-1}$. The grey shaded regions denote upstrokes, and one complete wingbeat cycle consists of the upstroke+downstroke.

- Phase 4, U-turn egress: After the apex of the turn, the bat begins to accelerate, descend, and straighten out the flight trajectory. During the egress, the bat continues to turn, but the radius of the turn increases rapidly. The U-turn egress consists of cycle 6.

Phase 1, the approach, consists of a deceleration from 1.93 m/s to 1.47 m/s, and an increase in angle of climb from 14 to 30 degrees. Simultaneously, the radius of curvature decreases from 200 cm to 60 cm (curvature change 0.5 to 1.7 m$^{-1}$) and the bearing angle changes from 15 degrees to 28 degrees. During phase 2, the initiation, the bat continues to climb and decelerate, but more notably rapidly decreases the radius of curvature to 7 cm (curvature of 14 m$^{-1}$) towards the end of this phase. The bat reaches its maximum height, minimum turn radius of 5.5 cm (curvature of 18 m$^{-1}$), and minimum velocity of 0.48 m/s during phase 3, the apex of the turn. Between the upstroke and downstroke of cycle 5 the bat is oriented in the $y_g$ – direction which is perpendicular to the long axis of the flight tunnel ($x_g$ – direction). Subsequently, the bat egresses the U-turn during phase 4, accelerating from 0.49 m/s to 0.71 m/s, increasing its radius of turn from 6 cm to 93 cm (curvature change 17 to 1), and begins a descent. By the

end of cycle 6, the bat is descending at an angle of 20 degrees and has dropped about 3 cm vertically ($z_g$ – direction) from the apex of the U-turn.

Results from a previously analyzed straight flight (Windes et al. [13]) are included in the present paper for comparison where appropriate. The straight flight data comes from the same *H. armiger* bat flying at approximate 2.0 m/s and was collected using the same experimental setup and computational analysis methods.

The following sections present analysis of the wing kinematics as well as analysis of the aerodynamic forces calculated using numerical flow simulation.

### 3.2 Wing kinematics analysis

Aerodynamic force asymmetries which allow the bat to maneuver are generated by the wing kinematics during the flight. Fig 4 provides a high level characterization of the wing kinematics based on the orientation of the stroke plane. The vertical and horizontal stroke plane angles capture the orientation of the total wing motion over each half-cycle. A more granular representation of the wing motion is provided in Fig 4. The motion of the wings relative to the stroke plane are characterized by the flap angle, stroke plane deviation angle, and the half-span for the 180 degree U-turn flight (a) and a straight flight (b) for comparison.—the flap angle, stroke plane deviation angle and half span capture the instantaneous position of the right and left wingtips relative to the stroke plane. The difference between the stroke plane angles (Fig 3) and the wing position parameters (Fig 4) can be described as kinematic trends across flap cycles versus trends within a single flap cycle.

During straight flight, the right and left wings exhibit a high degree of symmetry. The flap angle varies between -50 degrees to 40 degrees, the stroke plane deviation angle is bounded by -15 degrees during the downstroke and 20 degrees during the upstroke, and the half span

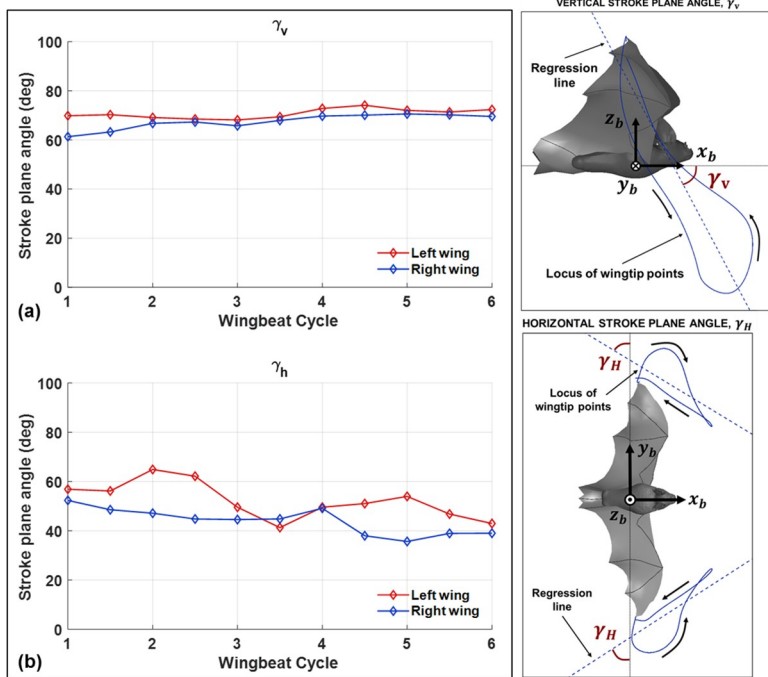

**Fig 3.** The vertical (a) and horizontal (b) stroke plane angle are shown for the right and left wing separately at each half-cycle as defined in the schematic above.

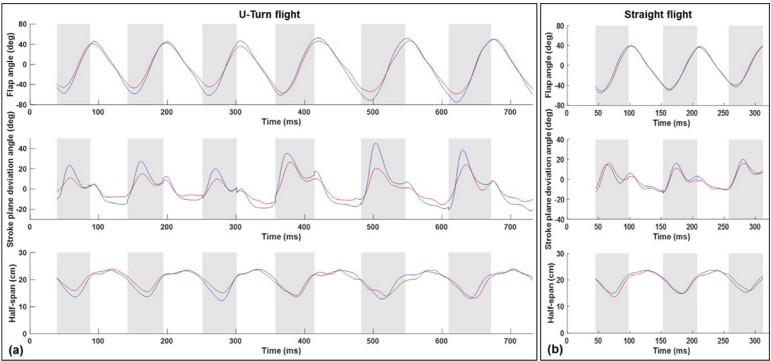

**Fig 4.** The motion of the wings relative to the stroke plane are characterized by the flap angle, stroke plane deviation angle, and the half-span for the 180 degree U-turn flight (a) and a straight flight (b) for comparison.

varies between a maximum value of approximately 25 cm to a minimum value of 14 cm during the upstroke when the wings are retracted towards the body. For the flight trajectory in this study, while relative symmetry exists in the vertical stroke plane angle, the horizontal stroke plane angle exhibits up to 30–40% difference between the right and left wings during portions of the flight. Specifically, the left wing on the outside of the turn has a higher angle relative to the right wing indicating that the right wing tip has stronger lateral movement. A similar trend was observed when analyzing a sweeping turn conducted by the same bat (Windes et al. [13]). In fact, during this flight, the right left asymmetry is not any greater during the U-turn (phases 2 to 4) when compared to the approach in phase 1 when the bat is gradually turning right. This indicates that modulation of the stroke plane orientation is not a major mechanism used by the bat to execute a tight U-turn compared to a gradual turn.

While the orientation of the stroke plane does not materially differ between gradual and tight turns, the wing motion relative to the stroke plane does differ significantly. During the initiation phase of the U-turn, the amplitude of the flap angle increases for both the right and left wings and the wingbeat frequency decreases simultaneously (Fig 5). Especially apparent in the frequency plot, there is a sharp decrease in frequency between the third and fourth wing-beat cycles. This correlates closely with both a decrease in flight velocity as well as a rapid decrease in radius of curvature of the turn (Fig 2). The decrease in radius of curvature can be equivalently described as an increase in angular velocity about the body-fixed $z_b$ – axis. A coordinated increase in flap amplitude and decrease in frequency allow the bat to maintain sufficient lift entering the U-turn. Throughout the flight, during both the approach and U-turn, the right wing to the inside of the turn exhibits a larger amplitude of flap angle relative to the left wing. This correlates with right/left asymmetry of the stroke plane deviation angle and is maximum during the early part of each upstroke. In the stroke plane deviation trend, a sharp change can be seen between cycles 1–3 and cycles 4–6. This indicates that the protracted motion of the right wing (inside the turn), which is a key marker of turning flight, increases proportionally to the tightness of the turn. The protracted motion is characterized by the wing tip exhibiting a larger downward and forward motion on the initiation of the upstroke than it normally does in level flight.

In order to further examine the wing kinematic asymmetry driving the U-turn, the trace of the wingtips and wrists are provided in Fig 6. Equivalent kinematics of straight flight is also shown for comparison. The dash lines represent the projection of the stroke plane and represent the vertical and horizontal stroke plane angles in the side and top view, respectively. The general trace of the wingtip in the top view ($z_b - y_b$) identifies the full span of the wing as the bat

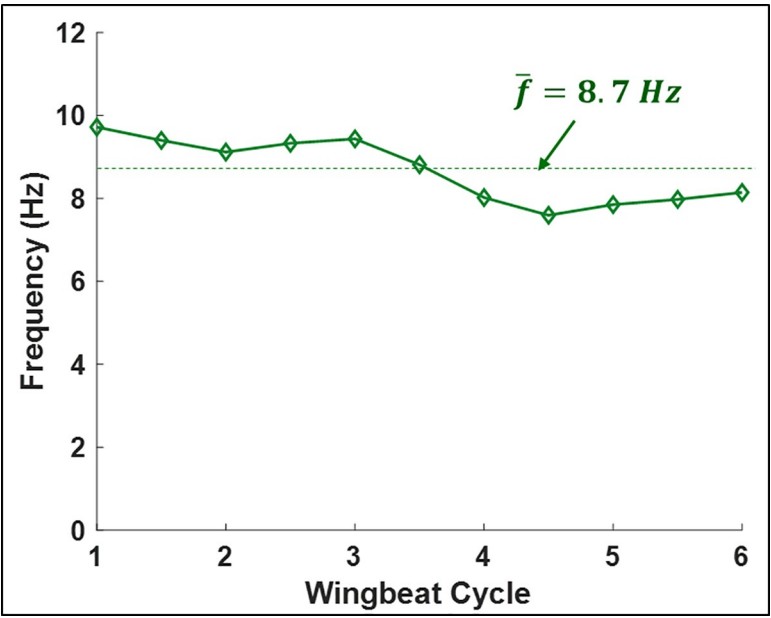

**Fig 5. Wingbeat frequency by half-stroke.** The mean frequency over the flight is 8.7 Hz.

stretches both wings out laterally during the downstroke, only to retract them inward towards the body on the upstroke. As shown in Fig 4, the motion of the wing in $z_b$−direction denoted by the flap amplitude varies between -50 degrees to 40 degrees for straight flight. The side view ($z_b$−$y_b$) shows that the wing sweeps back-to-front during the downstroke and front-to-back during the upstroke. As expected the straight flight exhibits a high degree of symmetry between the left and right wing motion. In contrast, during the gradual turning phase, modest but consistent right/left asymmetry can be observed throughout cycles 1–3; however, there is not much apparent temporal variation between the flaps. The asymmetry manifests as the right wing (inside of the turn) exhibiting a broader lateral traverse or larger stroke plane deviation. The inflection between the sweeping portion of the turn and the initiation of the U-turn occurs during the fourth cycle. During the fourth cycle not only does the right wing exhibit a large increase in stroke plane deviation as seen in the back view of the wingtip trace but also substantially increases the forward sweep of the wing during the upstroke as manifested in the side and top views. This is also accompanied by an increase in stroke plane deviation and forward sweep of the left wing but to a lesser extent than the right wing. As the bat approaches the apex of the turn in cycle 5, it retracts the right wingtip further towards it body during the upstroke resulting in still larger deviations from the stroke plane, while the left wing starts returning to its normal motion. This suggests that the left wing protraction may be marginally more impactful to the initiation of the U-turn while the right wing protraction is impactful to the execution of the U-turn. As the bat exits out of the U-turn in cycle-6, the left-right asymmetry still exists but returns to a state similar to that prior to cycles 4 and 5.

In summary, from the kinematics we see a sharp reduction in wingbeat frequency and increase in stroke amplitude as the bat initiates the U-turn. This change persists throughout the U-turn. The stroke plane angle data—i.e. small right/left asymmetry—suggest that asymmetries within each wingbeat cycle relative to the stroke plane drive the U-turn maneuver as opposed to changes in the mean orientation of the stroke plane. The larger flap amplitude and larger stroke plane deviation of the right wing (towards the inside of the turn) relative to the left wing exists throughout the initiation of the maneuver and increases during the apex phase.

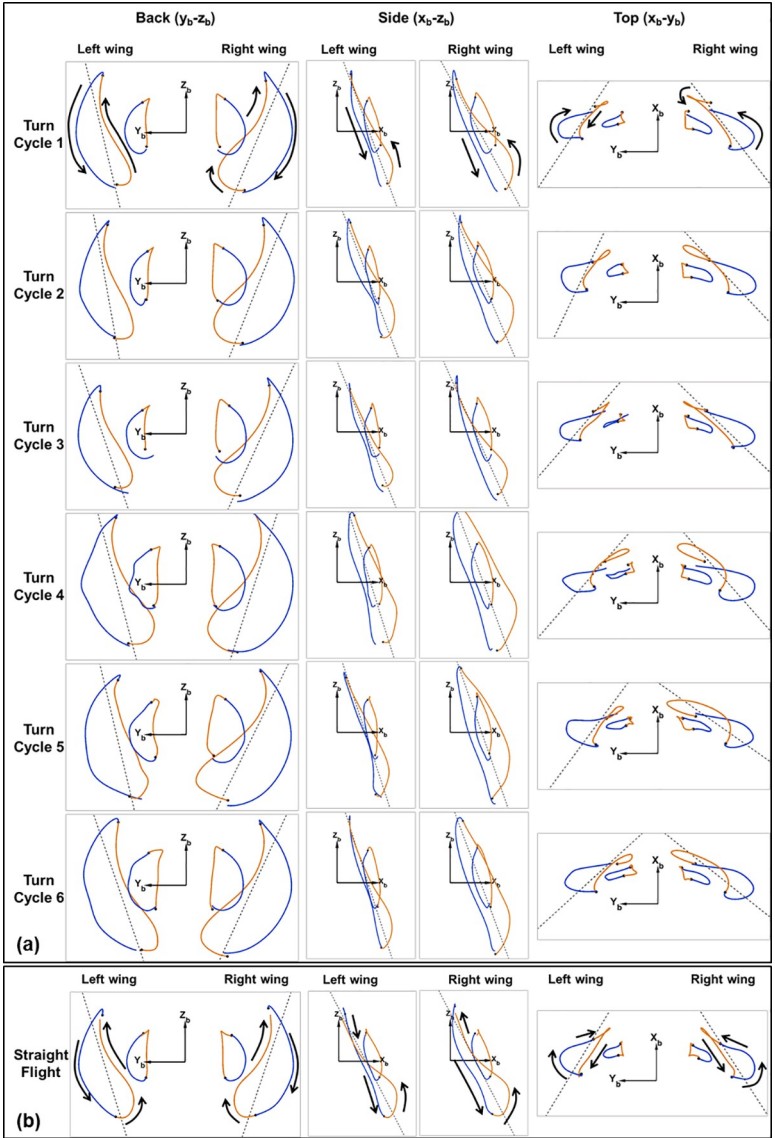

**Fig 6.** Trajectory of the wingtip and wrist for each wingbeat cycle of the U-turn shown in the body fixed coordinate system (a). A comparison from a representative straight flight is also provided (b).

From the wingtip traces, it can be seen that at the end of the downstroke and the beginning of the upstroke the right wing moves more laterally across the bat's body. The rotational inertia of wing can produce a counter rotation of the body into the turn to conserve angular momentum of the wing-body system. In order to further explore the turning mechanisms and to assess the relative importance of these observed inertia effects with aerodynamic mechanisms, we must investigate the aerodynamic forces generated by the wings during the maneuver.

### 3.3 Aerodynamic simulation results

Analysis of the aerodynamic forces during the U-turn allows us to examine how the wing kinematic asymmetries translate into turning forces and moments. Fig 7 shows the aerodynamic force coefficients throughout the flight in the body-fixed reference frame along with data from

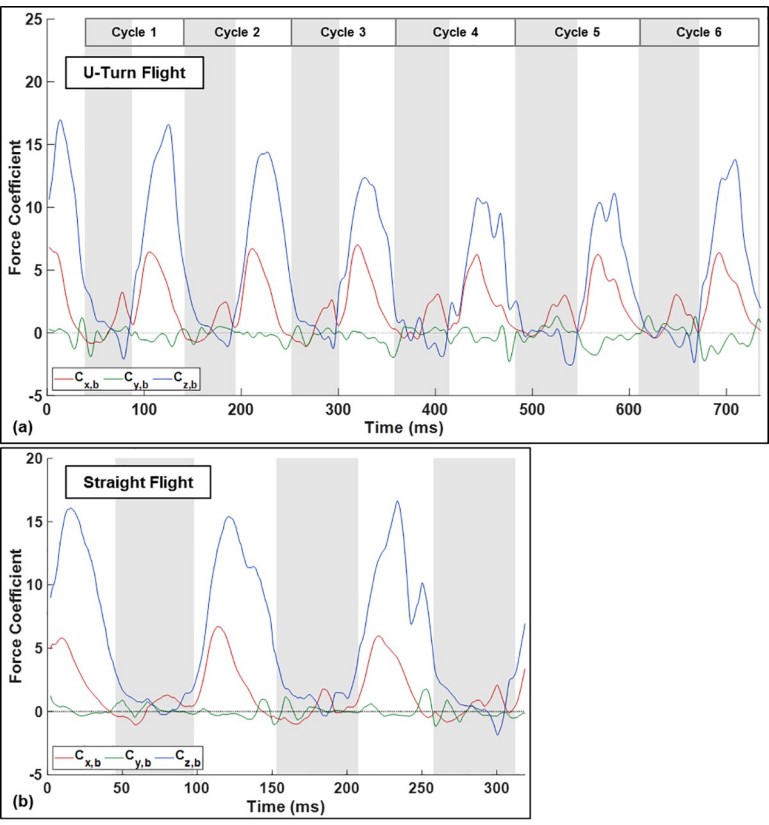

**Fig 7.** Force coefficients in the body frame for the U-turn flight (a). For comparison, results from a straight flight by the same bat is also provided (b).

a straight flight (Windes et al. [13]) as a baseline for comparison. The force coefficients, $C_x$, $C_y$, and $C_z$ are defined per standard convention as,

$$C = \frac{F}{\frac{1}{2}\rho U_\infty^2 A}$$

where $F$ is the directional force in the body-fixed system ($F_x$, $F_y$, or $F_z$), $\rho$ is the air density, $U_\infty$ is the initial approach flight velocity, and $A$ is the maximum planform area of the wing during the downstroke.

In both flight regimes (straight and U-turn), the lift force, responsible for keeping the bat aloft, has the largest magnitude of the three components and peaks during the mid-down-stroke. During the upstroke the bat avoids creating negative lift by retracting the wing inward towards its body and controlling the motion of the wing such that minimal fluid forces act on the wing. By doing so, in normal straight flight not only is negative lift avoided but also very little drag is induced during the upstroke–in fact some thrust is produced at the end of the upstroke. Upstroke thrust has been observed many times in bat flight as the flexible outer wing moves upward and backward during pronation to position for the downstroke; however, it is not universally present in all modes of flight. In spite of the small amount of thrust produced by the wings during the upstroke, seemingly the sole purpose of the upstroke in straight flight is to position the wing for the downstroke during which the bulk of lift and thrust is produced.

During the recorded turning flight, the magnitude of lift force is largest during the approach phase in cycle-1 and steadily decreases till it increases in magnitude after cycle-4 as it

initiates the tight U-turn and starts to ascend sharply. One important distinguishing factor from straight flight is the generation of negative lift towards the end of the upstroke in cycles 4 through 6 during the initiation and execution of the U-turn. While negative lift is produced during all six cycles, it is noticeably larger during cycle-4 and cycle-5. Not coincidental is the accompaniment of an increase in negative lift by an increase in positive thrust. Cycles 1–3 produce modest thrust (although noticeably larger than straight flight) near the conclusion of the upstroke and is accompanied by modest negative lift. This changes for cycles 4 and 5, which see relative larger magnitudes of thrust and negative lift during the upstroke. The bulk of thrust production occurs during the downstroke and peaks in value slightly earlier than the peak in lift. During turning flight, the magnitude of the thrust during the downstroke remains relatively constant for the first three cycles during the approach and initiation of U-turn, but increases in magnitude in cycle-4 and 5 during the U-turn and remains high in cycle-6. During the U-turn, upstroke thrust is accentuated and correlates with the decrease in radius of curvature. In order to achieve the elevated upstroke thrust, the bat allows larger negative lift.

The observed trends in kinematics and subsequent lift and thrust production point to the following scenario. Since the bat gains both height and vertical velocity during the approach and initiation of the U-turn, it down prioritizes lift during the tight portions of the U-turn in exchange for achieving rotation. In fact, we do see a decrease of vertical velocity, cresting, and subsequent decent during and after the apex of the U-turn clearly showing that the lift force drops below what would be required for maintaining altitude. This allows the bat to focus on generating excess lift entering the U-turn and subsequently prioritize rotation during the U-turn.

Contrary to lift and thrust, the lateral $y_b$ – force oscillates around zero and the cycle mean remains near zero throughout the flight. This mirrors observations from prior turning flights in which yawing (moment generation about $z_b$) and banking (tilting the lift force laterally) primarily drive the turn as opposed to a $y_b$ – force.

Validation of the computational framework for simulating bat flight has been conducted as a part of several of our prior studies (see [17–19]). However, in order to provide additional validation for this specific flight case, a mass dynamics analysis was run. The aerodynamic forces generated by the simulation were divided by the mass of the bat in order to get a predicted acceleration. The acceleration was integrated twice to obtain predicted position and velocity which is compared to the observed trajectory derived from the motion capture data. This is shown in Fig 8.

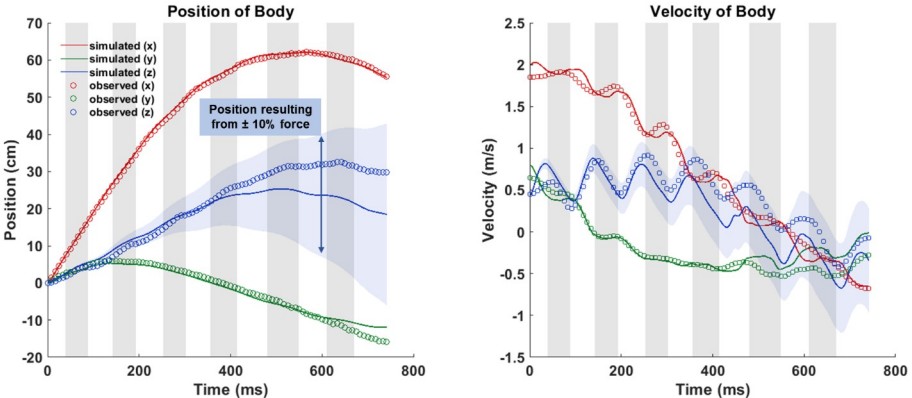

**Fig 8. Comparison of observed and predicted position and velocity of the bat body.**

Over the six wingbeat cycles, close agreement is seen in the global x and y-directions. The force in the z-direction is slightly under predicted but falls within a reasonable margin of error. Overlaid on the figure is an envelope containing the range of predicted z-position admitting to a possible ±10% over or under prediction of force. Due to the time integration, a constant force error accumulates substantially in the predicted position. Therefore, it is more appropriate to evaluate the predicted position relative to the envelop instead of the absolute deviation. Exact estimation of aerodynamic force is challenging in live flying animals and the ±10% envelop represents a reasonable margin of error based on what is typically achievable using a range of techniques [20].

In order to characterize the progress of the maneuver over time, Fig 9 provides the temporal evolution of rotational orientation in three-dimensional space. Three orientation angles—roll, pitch, and yaw—characterize the rotation of the bat's body in space. However, the instantaneous direction of flight as represented by the orientation of the velocity vector does not always exactly coincide with the body orientation as previously recognized by Iriarte-Diaz and Swartz [11] and Windes et al. [13]. The difference between the body orientation and velocity vector orientation is illustrated in two schematics in Fig 9.

Fig 9 provides the three body orientation angles, yaw (or "heading") angle, elevation angle (negative pitch), and the roll or bank angle. The velocity vector orientation is described by the bearing and climb angle. Broadly speaking, the bearing angle characterizes the turning component of the maneuver, while the climb angle describes the increase in height throughout the flight. Both of these angles are calculated based on the instantaneous orientation of the velocity vector relative to the ground reference frame. In contrast, the bank angle describes not the turn itself but rather the mechanism by which the turn is achieved. The yaw or heading angle when viewed in relation to the bearing angle indicates the deviation between the direction the bat is facing relative to the direction of the velocity vector. As expected, the trend in heading generally corresponds to the trend in bearing angle. That is, the bat nominally flies in the direction its body is facing. However, small differences between these angles describe whether changes in body orientation lead or lag the reorientation of the flight path.

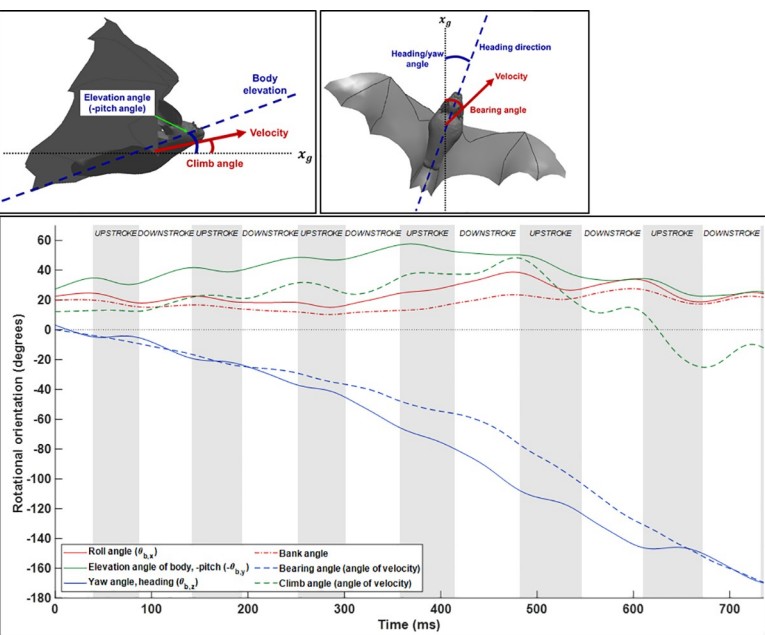

**Fig 9. Rotational orientation of the bat using both body-based angles (roll, elevation, and yaw) and velocity-based angles (bearing angle and climb angle).**

Throughout the approach, the difference between heading and bearing are negligible. Beginning in the initiation phase, the heading angle begins to change prior to change in the flight path. This suggests that the bat is using yaw rotation to reorient into the turn during the initiation phase of the U-turn. This closely mirrors observations seen by Iriate-Diaz and Swartz where turn initiation was mediated by yaw rotation [11]. The deviation between heading and bearing decreases during the apex of the U-turn and is eliminated during the egress. The trend between the heading and bearing angle shows that while the bearing angle undergoes a rapid change starting at cycle 4, the heading changes gradually over the whole U-turn maneuver. One explanation for why the heading change is more gradual over several flaps while the bearing change is comparatively more rapid, is that a change in the heading angle requires a yaw moment—the magnitude of the yaw moment is limited to some maximum value based on the upper limit of force asymmetry the wings can achieve. In contrast, the change in bearing angle can be partially achieved through tilting the net force vector by banking. Lift is the largest and most naturally generated force so simply redirecting this force through banking is less burdensome in comparison to generating large yaw moments.

The bat sustains a roll angle of approximately 20–25 degrees throughout the approach phase of the flight. This rapidly increases beginning at cycle 3 during the initiation phase of the U-turn to a final maximum of nearly 40 degrees. The peak roll angle is observed during end of the initiation phase and remains elevated throughout the apex before decreasing during egress. The increase in roll angle from 20 to 40 degrees closely corresponds to the point at which the bearing angle begins to change rapidly during cycle 4. Taking the heading, bearing, and roll angle together, we can conclude that the bat begins yaw rotation early in the U-turn which persists throughout the entire U-turn, while in contrast the change in bearing is related somewhat to the yaw but more significantly to the increase in roll angle.

The elevation angle describes the upward inclination angle of the bat's body while the climb angle describes the inclination of the velocity vector. Prior studies have indicated that the elevation angle is persistently higher than the climb angle during all modes of flight (Windes et al. [13]). That is, the bat naturally flies with a slight body inclination of approximately 20 degrees during level flight. In the present flight, during the approach the bat initially climbs at 10 degrees increasing up to 30 degrees by the beginning of the initiation of the U-turn. The elevation angle is uniformly 20 degrees higher than the climb angle throughout the approach. During the initiation of the U-turn, the climb angle continues increasing from 30 degrees to the maximum of 43 degrees. Subsequently throughout the apex and egress, the climb angle rapidly decreases reaching zero (horizontal relative to the ground frame) in cycle 6. During the egress, the bat begins to descend (i.e. negative climb angle); however, the elevation angle of the body remains positive. One key result of this is that the lift force vector remains more vertically oriented. During decent, the bat is able to leverage gravitational force to accelerate and is not incentivized to use a large thrust force to achieve this.

Examination of the bat's body and velocity orientation angles suggests that rolling is responsible for a large portion of the bearing angle change during the U-turn. In order to further investigate this phenomenon, Fig 10 shows the net force in components of tangential, radial, and vertical force defined relative to the trajectory of the flight. Tangential force acts to accelerate or decelerate the bat along the trajectory, radial force acts perpendicular to the tangent of the trajectory and serves to change the bearing angle, and the vertical force opposes gravity keeping the bat aloft or allowing the bat to climb.

The vertical force, although defined relative to the ground vertical axis, shows a similar trend to the body frame lift force. That is, when the bat begins to climb in the approach phase the vertical force is maximum. As the rate of height gain decreases throughout the initiation and apex phases, the vertical force decreases. As the bat egresses the U-turn, it arrests its rate of

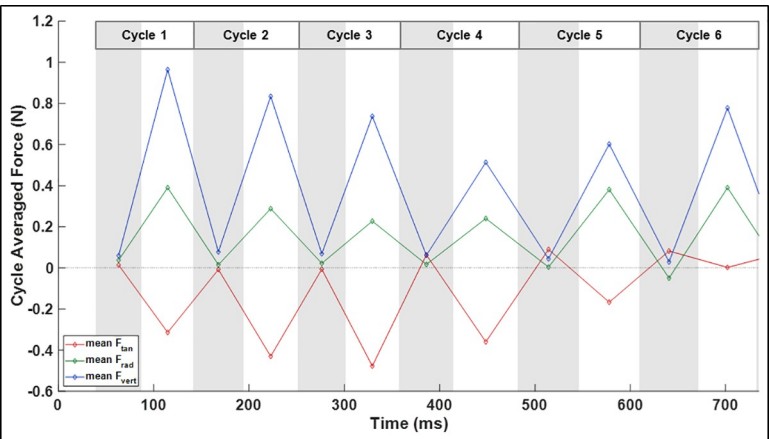

**Fig 10. Tangential, radial, and vertical components of aerodynamic force relative to the flight trajectory.**

descent by slightly beginning to increase the vertical force to a magnitude which supports its weight. This serves to arrest its descent towards a more gradual downward trajectory preventing a precipitous loss of altitude. The tangential force remains negative over the first four cycles corresponding to the observed deceleration. This is critical to maneuvering a tight turn since it decreases the required centripetal force to achieve a given radius of curvature. For a bat of a given mass to achieve a turn of a particular radius of curvature, the radial force requirement increases with the square of the tangential velocity. That is, to achieve a turn with a radius of 6 cm, a bat flying at 1 m/s would need achieve 4 times the radial force compared to a 0.5 m/s tangential flight velocity. For the present flight, the negative tangential force causing deceleration is minimum during the initiation phase of the U-turn, specifically during the third wingbeat cycle.

The radial force causes a centripetal acceleration of the bat causing a change in trajectory of the flight. This centripetal acceleration directly causes the change in bearing angle shown in Fig 9. While it may be expected that the radial force would be maximum during the apex of the turn, it is critical to keep in mind the large impact that flight velocity has on the required radial force. Since the bat is decelerating throughout the initiation of the turn, the required radial force actually decreases substantially during the apex of the U-turn relative to the prior cycles. This is one additional explanation for why the lift force decreases during the apex of the U-turn—the lift force vector which is tilted radially by the roll angle does not need to be as high to turn the bat given the lower flight velocity.

The radial force as shown in Fig 10 explains the change in bearing angle and flight trajectory; however in order to investigate the change in body orientation—roll and heading—we need to view the action of rotational moments on the bat which re-orient the body. The aerodynamic force moments relative to the estimated center of mass of the bat are shown in Fig 11A. Moments on the bat induce angular accelerations which change the angular velocity of the bat's body. Thus, the angular acceleration is provided as well in Fig 11B. The center of mass for the purpose of calculating the moments was approximated as a fixed point on the middle of the bat's body. The aerodynamic moments and the angular accelerations are related by the moment of inertia of the body-wing system which is time-varying as the wings flap. While the trends in moment and angular acceleration should be similar, the plots should not be expected to match up completely since the moment of inertia varies temporally.

Explanation of the moments is best conducted by observing general trends in the plots. This is because there are a few factors which contribute to the challenge of interpreting rotational moments in a very specific time-dependent manner. First, as the wings flap the location

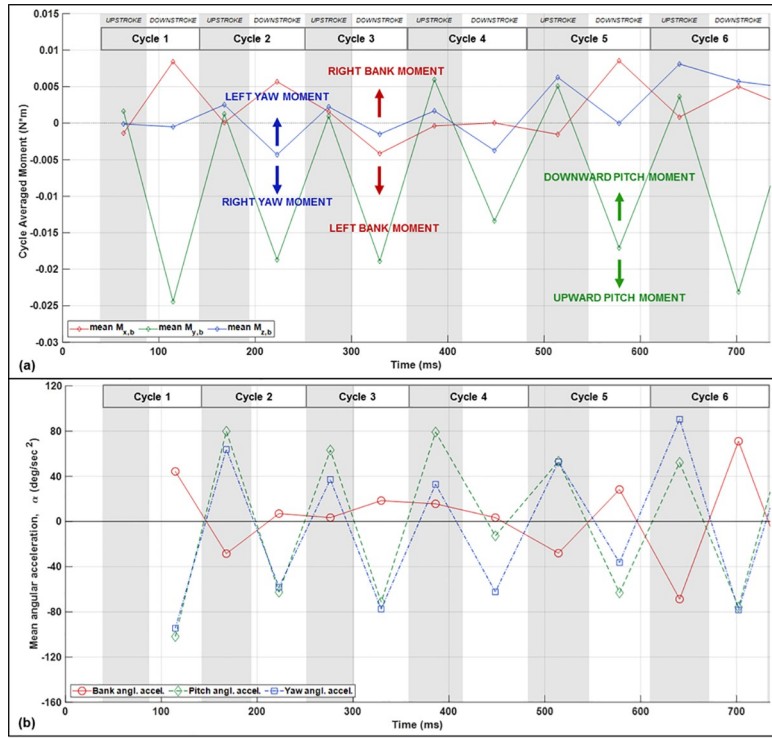

**Fig 11.** Half-cycle mean aerodynamic moments relative to the approximate center of mass calculated from the aerodynamic forces (a). Half-cycle mean angular acceleration of the bat body defined as rate of change of rotational velocity about the body-fixed coordinate axes (b).

of the center of mass shifts thus the moments shown are only an estimate of the precise rotational moment. Second, the rotational moments are derived only from the aerodynamic forces. In addition, inertial moments may also contribute to reorienting the bat's body. Thus, the aerodynamic moments are not guaranteed to explain 100% of the body rotation. However, despite these caveats, the trends in the aerodynamic moments are largely consistent with the observed changes in angular orientation. Additionally, comparing the aerodynamic moments and the angular acceleration can provide insight as to the relative contribution of aerodynamic mediated rotation versus inertia mediated rotation.

Negative yaw moment (right yaw) is observed during the downstroke of cycles 2–4 during the period of the turn when yaw angular velocity is increasing. Subsequently, during the later portion of the apex and moving into the egress, the rate of yaw rotation decreases and correspondingly the yaw moment becomes positive. This trend in the yaw moment corresponds to the trend observed in the angular acceleration. In addition, both yaw moment and yaw acceleration vary substantially between the upstoke and downstroke; during the downstroke, yaw moment and acceleration is directed into the turn while counter rotation is observed during the upstrokes. During the initiation, the magnitude of the right yaw during the downstroke is greater than the left yaw during the upstroke resulting in a net right rotation. The yaw rotational acceleration is slowed during the apex and egress of the U-turn. It is important to recognize that the yaw moment and angular acceleration describe the rate of change of the angular velocity—thus despite the rate of yaw rotation decreasing during the apex and egress, the yaw angular velocity maintains a right rotation through the duration of the flight. That is, the rate of right yaw rotation slows during the later portion of the maneuver but right rotation persists throughout the maneuver.

Both the roll and yaw moments are highly variable between the upstroke and downstroke. Generally, for a right turn, right roll and right yaw are maximum during each downstroke and reverse during each upstroke. This provides two available mechanisms for increasing net moment across a complete wingbeat cycle. Either the downstroke moment may be elevated or the reverse moment during the upstroke may be mitigated. Thus despite the majority of the right roll and right yaw moment generation during the downstroke, modulation of the moment during the upstroke will contribute to the net moment experienced over the course of a full wingbeat cycle. An additional consideration is that roll rotation may be mediated by inertial effects. Roll moment is positive during downstrokes 1 and 2, indicating right roll; however, throughout the initiation phase (cycles 3–4), the aerodynamic roll moment does not appear to fully explain the roll angular acceleration shown in Fig 11B and the orientation shown in Fig 9.

In order to examine the source and nature of the aerodynamic moments at various points in the wingbeat cycle, the aerodynamic force in the body frame is broken up by the region of the wing on which it acts and is shown in Fig 12. Nominally, asymmetry in the $x_b$-force will cause yaw moments while asymmetry in the $z_b$-force will cause roll moments.

The thrust on the outer left wing relative to the inner left wing is elevated throughout the downstrokes of the initiation phase of the turn. In general, thrust generated by the outer wing will have a disproportionally larger contribution to aerodynamic moment relative to the inner wing since the moment arm is longer. The wingspan of around 50 cm is substantially longer than other dimensions of the bat such as the mean wing chord or the body length, so small asymmetries on the outer wing can have a large rotational effect. This appears to contribute to the yaw moment generated throughout the approach and initiation phase of the turn.

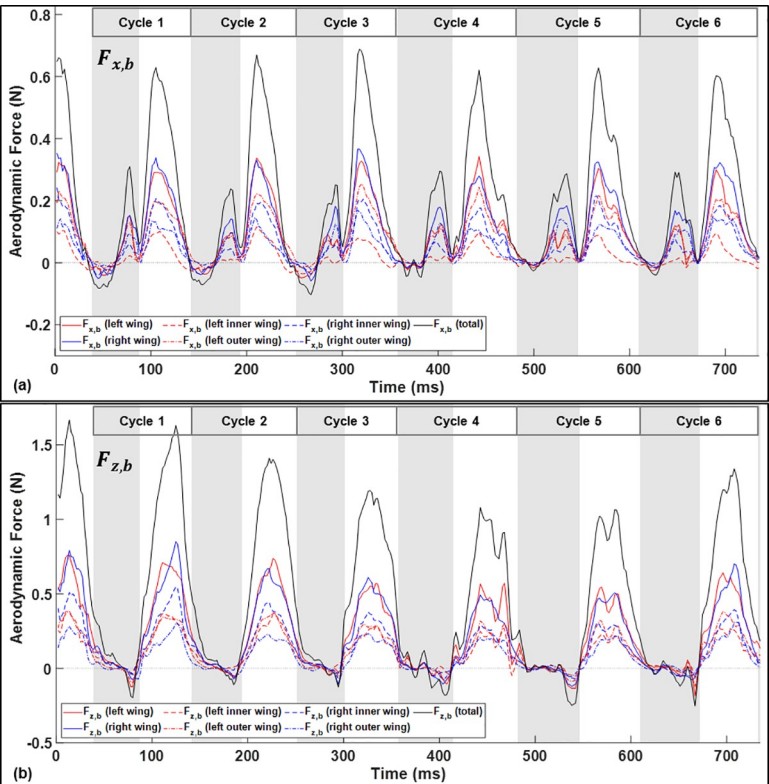

**Fig 12.** Thrust force (a) and lift force (b) calculated in the body-fixed frame partitioned by where the force is acting. Solid lines represent right and left wing totals. The dashed and dashed-dotted lines represent the force acting on the outer and inner right and left wings.

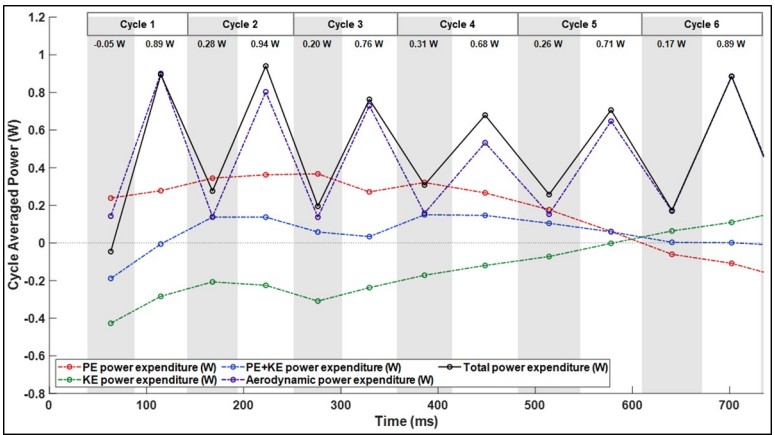

**Fig 13. Total power expenditure defined as the sum of aerodynamic power, kinetic energy expenditure, and potential energy expenditure.**

Lift force is overwhelming generated during the downstroke of each cycle. This aligns with the observation that right roll moment is generated almost exclusively during the downstrokes. Asymmetry in $F_{z,b}$ between the right and left wings is marginal throughout the flight; however, the specific location on the wing at which the force acts will change the length of the moment arm. During downstrokes 2–3, the lift generated on the right outer wing is depressed causing a net right roll moment corresponding to the increase in bank angle which occurs throughout the initiation phase of the turn.

## 3.4 Power and energy analysis

In order to investigate the power requirement of a tight U-turn maneuver, total power expenditure was calculated over the entire flight. Total power (Fig 13) is the sum of aerodynamic power, rate of change in potential energy (resulting from height change), and rate of change in kinetic energy (resulting from velocity change). Table 2 provides the power expenditure calculated for the U-turn as well as comparison to two other fights—a straight flight and a sweeping turn.

Total power is elevated throughout the approach (cycles 1–2) corresponding with the time at which the bat is initiating its climb. This agrees with prior data which indicated that vertical height gain is a power intensive mode of flight. Total power expenditure remains elevated

**Table 2. Mean total power expenditure for a straight flight, sweeping turn [13], and U-turn.**

| POWER | | Cycle mean (Watts) | times straight flight | Upstroke mean | times straight flight | Downstroke mean | times straight flight |
|---|---|---|---|---|---|---|---|
| Straight flight | | 0.34 | - | 0.01 | - | 0.68 | - |
| Sweeping turn [13] | Initiation Climb/Turn | 0.66 | 1.94 | 0.28 | 28.0 | 1.03 | 1.51 |
| | | 0.58 | 1.71 | 0.16 | 16.0 | 0.99 | 1.46 |
| | | 0.74 | 2.18 | 0.37 | 37.0 | 1.10 | 1.62 |
| U-turn | Approach Initiation Apex Egress | 0.51 | 1.50 | 0.20 | 20.0 | 0.83 | 1.22 |
| | | 0.52 | 1.53 | 0.12 | 12.0 | 0.92 | 1.35 |
| | | 0.49 | 1.44 | 0.26 | 26.0 | 0.72 | 1.06 |
| | | 0.49 | 1.44 | 0.26 | 26.0 | 0.71 | 1.04 |
| | | 0.58 | 1.71 | 0.17 | 17.0 | 0.98 | 1.44 |

throughout the maneuver when compared to the cost of straight flight. At the apex of the U-turn, the cycle mean power expenditure of 0.49 W is about 1.44 times higher than what has been previously calculated for the same bat in straight flight (cycle mean of 0.34 W) as shown in Table 2. This suggest that even when the bat crests and ceases to climb, the cost of executing the U-turn is materially elevated relative to the straight flight baseline. Notably, the primary elevation in power relative to straight flight is observed during the upstrokes. This suggests that the bat is expending energy to generate elevated thrust and consequently yaw moments during the upstroke. Somewhat surprisingly, the maximum power expenditure (by a small margin) is observed during the egress of the turn. During the egress, both the downstroke thrust is maximum (Fig 7) and the tangential force becomes positive (Fig 10). Thus we can assign the primary power cost during the egress to acceleration out of the turn. While it is true that gravity aids the acceleration process, the observed positive tangential aerodynamic force indicates that thrust and gravitational force are used in concert.

## 4. Conclusion

Analysis of the wing kinematics in conjunction with aerodynamic parameters during each of the four phases of the flight provides several insights into the mechanisms the bat is using to execute a U-turn maneuver. As the bat climbs and decelerates during the initiation of the turn, the wingbeat frequency decreases and the flap amplitude increases. The right wing on the inside of the turn exhibits a greater flap amplitude compared to the left wing, and extends across the body toward the center axis of the body during the end of the downstroke and beginning of the upstroke. This phenomenon can be clearly seen in the wingtip locus visualization and manifests as a very high positive stroke plane deviation angle in the right wing during the initiation and apex of the U-turn. Lateral movement of the right wing causes an inertial effect causing counter rotation of the body to conserve angular moment. However, the presence of aerodynamic yaw moment suggests that the yaw rotation is not simply an inertia rotation but a combined aerodynamic and inertial mechanism.

The bat generates steady modest yaw rotation throughout the approach and initiation of the U-turn during both the upstroke and downstroke to reorient the body. Throughout the initiation phase the bank angle nearly doubles from 20 to 40 degrees causing a large lateral force. This causes the bearing angle to change rapidly throughout the fourth and fifth cycles eventually catching up to the more gradually changing yaw angle in the sixth cycle. In addition to the high roll angle, the slower flight velocity allows the lateral force to more quickly turn the bat since the required centripetal force is lowered.

From this analysis, we can make several high level observations about the U-turn flight. The bat's ability to rapidly yaw appears to be limited to a degree so the yaw rotation began about one to two cycles before the rapid bearing angle change and was stretched out over many wingbeat cycles. During the apex of the turn, the bat combined a high roll angle with a low flight velocity magnitude to very rapidly redirect its bearing direction and negotiate a low radius of curvature flight trajectory. Right roll moments and right yaw moments were primarily generated during the down stroke phase of the wingbeat cycle. The general trends in aerodynamic moment and angular acceleration of the body were correlated, but they did not precisely match on an instantaneous basis suggesting a shared contribution of inertial and aerodynamic mediated rotation mechanisms.

In the future, further investigation of maneuvers executed by different bat species at different flight velocities and turning radii will allow a more generalized understanding of how bats turn. We believe this study highlights the importance of simultaneous kinematic and aerodynamic analysis in order to understanding the turning mechanisms being employed. A general

model of how various different bat flight maneuvers are executed would be a valuable tool for both understanding bat biomechanics and enabling novel design of bat-mimetic micro air vehicles.

## Supporting information

**S1 Animation. Aerodynamic simulation results.** Animation of the U-turn maneuver using results from the numerical aerodynamic simulation. Isosurfaces of coherent vorticity are shown along with contours of pressure coefficient differential on the wing surface (pressure below the wing minus pressure above the wing).
(MP4)

**S1 Video frames. Time series of frames from the bat flight video, camera 1.** A time series of video frames of the bat executing the u-turn maneuver are provided. The video was recorded at 120 frames per second, so the time separating each subsequent frame is 8.33 ms.
(RAR)

**S2 Video frames. Time series of frames from the bat flight video, camera 2.** A time series of video frames of the bat from a second camera perspective complementing S1 Video frames. Since the cameras were temporally synchronized, corresponding frames represent the same point in time.
(RAR)

## Acknowledgments

The authors thank Advanced Research Computing at Virginia Tech for providing the computational resources which made this research possible (URL: http://www.arc.vt.edu). The authors also thank collaborators Susheel Sekhar, Matt Bender, Aevelina Rahman, Josh Lesser, Yang Xu, Yuxian Ye, Mengfan Wang, Junyang Xu, Han Xu, and Xuchen Gu.

## Author Contributions

**Conceptualization:** Peter Windes, Danesh K. Tafti, Rolf Müller.

**Data curation:** Peter Windes.

**Formal analysis:** Peter Windes.

**Funding acquisition:** Danesh K. Tafti.

**Investigation:** Peter Windes, Danesh K. Tafti.

**Methodology:** Peter Windes, Danesh K. Tafti, Rolf Müller.

**Project administration:** Rolf Müller.

**Resources:** Danesh K. Tafti, Rolf Müller.

**Software:** Peter Windes, Danesh K. Tafti.

**Supervision:** Danesh K. Tafti, Rolf Müller.

**Validation:** Peter Windes.

**Visualization:** Peter Windes.

**Writing – original draft:** Peter Windes.

**Writing – review & editing:** Peter Windes, Danesh K. Tafti, Rolf Müller.

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
