## [Decision Letter · Decision Letter 0]

4 Aug 2020

PONE-D-20-20383

Analysis of a 180-degree U-turn maneuver executed by a Hipposiderid bat

PLOS ONE

Dear Dr. Windes,

Thank you for submitting your manuscript to PLOS ONE. After careful consideration, we feel that it has merit but does not fully meet PLOS ONE’s publication criteria as it currently stands. Therefore, we invite you to submit a revised version of the manuscript that addresses the points raised during the review process.

We look forward to receiving your revised manuscript.

Kind regards,

Fang-Bao Tian

Academic Editor

PLOS ONE

Journal Requirements:

2. In your Methods section, please provide additional details regarding the animals used in your study and ensure you have described the source. For more information regarding PLOS' policy on materials sharing and reporting, see https://journals.plos.org/plosone/s/materials-and-software-sharing#loc-sharing-materials.

3. In your Methods section, please include a comment about the state of the animals following this research. Were they released or housed for use in further research?

4. Thank you for including your ethics statement:  "Ethical bat housing and handling guidelines specified by Virginia Tech’s Institutional Animal Care and Use Committee (IACUC protocol number 15-067) were followed throughout the study".   

Please amend your current ethics statement to confirm that your named ethics committee specifically approved this study.

For additional information about PLOS ONE submissions requirements for ethics oversight of animal work, please refer to http://journals.plos.org/plosone/s/submission-guidelines#loc-animal-research  

5. We note you have included a table to which you do not refer in the text of your manuscript. Please ensure that you refer to Table 2 in your text; if accepted, production will need this reference to link the reader to the Table.

Reviewers' comments:

Reviewer's Responses to Questions

**Comments to the Author**

1. Is the manuscript technically sound, and do the data support the conclusions?

Reviewer #1: Yes

2. Has the statistical analysis been performed appropriately and rigorously? 

Reviewer #1: Yes

3. Have the authors made all data underlying the findings in their manuscript fully available?

Reviewer #1: Yes

4. Is the manuscript presented in an intelligible fashion and written in standard English?

Reviewer #1: Yes

5. Review Comments to the Author

Reviewer #1: The authors used 3D optical motion capture and aerodynamic simulations to investigate a U-turn maneuver executed by a great roundleaf bat.

Over the course of the entire flight, the bat first begins to gradually turn and climb, then executes a tight U-turn, and lastly descends while exiting the turn. This sequence was categorized into four phases. Analysis of the wing kinematics in conjunction with aerodynamic parameters during each of the four phases of the flight provides several insights into the mechanisms the bat is using to execute a U-turn maneuver.

The topic is important and within the scope of PLOS ONE. The manuscript is well written.

I would like to recommend it for publication. However, I think the following suggestion would be helpful to improve the quality of the manuscript.

1. The authors pointed that "certain unsteady mechanisms such as enhanced lift by the leading edge vortex (LEV) have been shown to be nearly ubiquitous in flapping flight, while other mechanisms such as clap and fling are only observed in certain insects operating at smaller length scales" on page 9. I think it is better to also point that bats can dynamically changing wingspan to enhance lift (Wang et al. J R Soc Interface, 12(113), 20150821,2015; Physics of Fluids, 26(6), 061903, 2014; Physics of Fluids, 27(6), 061901, 2015)

2. The authors use 44 cells per wing chord length in the simulations. The required number of cells in numerical simulations depends strongly on the Reynolds number of the flows. What is the typical Reynolds number in the simulations?

3. What is the initial flow condition for the simulations?

4. I suggest the authors to provide the details of the definition of force coefficients in Figure 7.

6. PLOS authors have the option to publish the peer review history of their article (what does this mean?). If published, this will include your full peer review and any attached files.

Reviewer #1: No

While revising your submission, please upload your figure files to the Preflight Analysis and Conversion Engine (PACE) digital diagnostic tool, https://pacev2.apexcovantage.com/. PACE helps ensure that figures meet PLOS requirements. To use PACE, you must first register as a user. Registration is free. Then, login and navigate to the UPLOAD tab, where you will find detailed instructions on how to use the tool. If you encounter any issues or have any questions when using PACE, please email PLOS at figures@plos.org. Please note that [Sec sec009] files do not need this step.

---

## [Author Response · Author response to Decision Letter 0]

11 Oct 2020

1. The authors pointed that "certain unsteady mechanisms such as enhanced lift by the leading edge vortex (LEV) have been shown to be nearly ubiquitous in flapping flight, while other mechanisms such as clap and fling are only observed in certain insects operating at smaller length scales" on page 9. I think it is better to also point that bats can dynamically changing wingspan to enhance lift (Wang et al. J R Soc Interface, 12(113), 20150821,2015; Physics of Fluids, 26(6), 061903, 2014; Physics of Fluids, 27(6), 061901, 2015)

Author Reply: We have added the reference per the suggestion.

2. The authors use 44 cells per wing chord length in the simulations. The required number of cells in numerical simulations depends strongly on the Reynolds number of the flows. What is the typical Reynolds number in the simulations?

Author Reply: The Reynolds number varies with flight velocity (as the bat is slowing down during the u-turn), but ranges from 4,000 to 10,000. We have mentioned this in the text.

3. What is the initial flow condition for the simulations?

Author Reply: The initial flow condition is zero absolute velocity (since the bat is flying in stagnant air in the flight tunnel), and we allow several time steps for the flow to develop. Since the reference frame is moving, the velocity field was initialized at the reference frame velocity. We have mentioned this in the text.

4. I suggest the authors to provide the details of the definition of force coefficients in Figure 7.

Author Reply: We have added this definition.

---

## [Editor Report · Decision Letter 1]

13 Oct 2020

PONE-D-20-20383R1

Analysis of a 180-degree U-turn maneuver executed by a Hipposiderid bat

PLOS ONE

Dear Dr. Windes,

Thank you for submitting your manuscript to PLOS ONE. After careful consideration, we feel that it has merit but does not fully meet PLOS ONE’s publication criteria as it currently stands. Therefore, we invite you to submit a revised version of the manuscript that addresses the points raised during the review process.

We look forward to receiving your revised manuscript.

Kind regards,

Fang-Bao Tian

Academic Editor

PLOS ONE

Additional Editor Comments (if provided):

Thank you for revising your work. I have checked it, and would like to request a minor revision regarding the definitions of Reynolds number and lift coefficient. In 3.3, the lift coefficient is defined by using "the mean flight velocity". I think this reference velocity can be used in the definition of Reynolds number. By the way, it is better to use "lift coefficient" instead of "force coefficient" in 3.3.

---

## [Author Response · Author response to Decision Letter 1]

15 Oct 2020

please see the file author response2.docx

---

## [Editor Report · Decision Letter 2]

16 Oct 2020

Analysis of a 180-degree U-turn maneuver executed by a Hipposiderid bat

PONE-D-20-20383R2

Dear Dr. Tafti,

We’re pleased to inform you that your manuscript has been judged scientifically suitable for publication and will be formally accepted for publication once it meets all outstanding technical requirements.

Kind regards,

Fang-Bao Tian

Academic Editor

PLOS ONE
---

## [Editor Report · Acceptance letter]

23 Oct 2020

PONE-D-20-20383R2 

Analysis of a 180-degree U-turn maneuver executed by a Hipposiderid bat 

Dear Dr. Tafti:

I'm pleased to inform you that your manuscript has been deemed suitable for publication in PLOS ONE. Congratulations! Your manuscript is now with our production department. 

Kind regards, 

on behalf of

Dr. Fang-Bao Tian 

Academic Editor

PLOS ONE